# Patterns of Alcohol Consumption and Use of Health Services in Spanish University Students: UniHcos Project

**DOI:** 10.3390/ijerph19106158

**Published:** 2022-05-18

**Authors:** Esperanza Romero-Rodríguez, Carmen Amezcua-Prieto, María Morales Suárez-Varela, Carlos Ayán-Pérez, Ramona Mateos-Campos, Vicente Martín-Sánchez, Rocío Ortíz-Moncada, Susana Redondo-Martín, Juan Alguacil Ojeda, Miguel Delgado-Rodríguez, Gemma Blázquez Abellán, Jéssica Alonso-Molero, José María Cancela-Carral, Luis Félix Valero Juan, Tania Fernández-Villa

**Affiliations:** 1Maimonides Biomedical Research Institute, Reina Sofia University Hospital, University of Córdoba, 14001 Cordoba, Spain; emromerorodriguez@gmail.com; 2Consortium for Biomedical Research in Epidemiology & Public Health (CIBER Epidemiología y Salud Pública-CIBERESP), 28029 Madrid, Spain; carmezcua@ugr.es (C.A.-P.); maria.m.morales@uv.es (M.M.S.-V.); vicente.martin@unileon.es (V.M.-S.); mdelgado@ujaen.es (M.D.-R.); 3Department of Preventive Medicine and Public Health, University of Granada, 18016 Granada, Spain; 4Department of Preventive Medicine and Public Health, Food Sciences, Toxicology and Legal Medicine, University of Valencia, 46100 Valencia, Spain; 5Well-Move Research Group, Department of Special Didactics, University of Vigo, 36310 Vigo, Spain; cayan@uvigo.es; 6Area of Preventive Medicine and Public Health, University of Salamanca, 37007 Salamanca, Spain; rmateos@usal.es (R.M.-C.); luva@usal.es (L.F.V.J.); 7Department of Biomedical Sciences, University of León, 24071 León, Spain; 8The Research Group in Gene-Environment and Health Interactions, Institute of Biomedicine (IBIOMED), University of León, 24071 León, Spain; 9Public Health Research Group, Food and Nutrition Research Group, University of Alicante, 03550 Alicante, Spain; rocio.ortiz@ua.es; 10Department of Preventive Medicine and Public Health, University of Valladolid, 47005 Valladolid, Spain; susana.redondo@uva.es; 11Department of Clinical and Experimental Psychology, University of Huelva, 21071 Huelva, Spain; alguacil@dbasp.uhu.es; 12Department of Health Sciences, University of Jaén, 23071 Jaén, Spain; 13Department of Medical Sciences, School of Pharmacy, University of Castilla-La Mancha, 02008 Albacete, Spain; gemma.blazquez@uclm.es; 14Department of Preventive Medicine and Public, University of Cantabria-IDIVAL, 39008 Santander, Spain; alonsomoleroj@gmail.com; 15Galicia Sur Health Research Institute (IIS Galicia Sur), SERGAS-UVIGO, 36312 Vigo, Spain; chemacc@uvigo.es; 16Department of Specials Didactics, University of Vigo, 36310 Vigo, Spain

**Keywords:** alcohol drinking, alcohol consumption patterns, university student, health services research, emergency services, primary care, cross-sectional studies

## Abstract

The aim of the study was to examine the association of alcohol consumption patterns (hazardous alcohol use and binge drinking) and the use of emergency services and primary care consultations in university students. An observational, descriptive, cross-sectional study was conducted at eleven Spanish universities collaborating within the uniHcos Project. University students completed an online questionnaire that assessed hazardous alcohol use and binge drinking using the AUDIT questionnaire and evaluated the use of emergency services and primary care. A descriptive analysis of the data was performed, as well as the chi-squared test and Student’s *t*-test and nonconditional logistic regression models to examine this association. Results: There were 10,167 participants who completed the questionnaire. The prevalence of hazardous alcohol use was 16.9% (95% CI: 16.2–17.6), while the prevalence of binge drinking was 48.8% (95% CI: 47.9–49.8). There were significant differences in the use of emergency services in those surveyed with hazardous alcohol use (*p* < 0.001) or binge drinking pattern (*p* < 0.001). However, no significant differences were observed in terms of attendance during primary care visits in individuals with hazardous alcohol use (*p* = 0.367) or binge drinking pattern (*p* = 0.755). The current study shows the association between university students with a pattern of hazardous alcohol use or binge drinking and greater use of emergency services. However, no significant association was observed between the said consumption patterns and the use of primary care services.

## 1. Introduction

Alcohol consumption is one of the main preventable causes of morbidity and mortality [1]. The European Union has the highest rate of alcohol consumption in the world, meaning there is a greater burden of disease and death on this continent due to the consumption of this substance [2,3]. In Spain, alcohol is the most widely used psychoactive substance, followed by tobacco and cannabis. Although alcohol intake has decreased slightly since 2015 in Spain [4], its consumption continues to be a public health problem in adolescents and young adults.

Identifying toxic habits, such as alcohol consumption, in adolescence and youth is of vital importance since their presence at an early age carries significant consequences for development in adulthood [5]. In many cases, entering university brings changes in lifestyle, marked by a change in housing, a difference in the demands of university education, or by becoming part of a new social network [6].

Various international studies have addressed patterns of alcohol consumption in university students [7,8,9,10,11,12,13,14]. The prevalence of hazardous alcohol use and binge drinking varies in international series, although there is a significant increase in binge drinking in this population. Research consistently shows that young people tend to drink more in late adolescence and early adulthood [12,13], with young adults especially prone to binge drinking [14]. Therefore, in addition to identifying alcohol consumption in this population, it is necessary to establish the pattern of the said consumption, since the type of alcohol intake is decisive in developing alcohol-related problems [14].

Alcohol consumption leads to a significant increase in the use of health services, both at the hospital level and in primary care (PC). Although the health repercussions of alcohol consumption on the individual and on society are known and the literature shows that PC and emergency services are important points in detecting alcohol consumption and assessing the need to initiate quick interventions [15,16,17,18], the evidence on the relationship between drinking patterns and health care use is not well-established. Some studies have described that patients with alcohol dependence have a greater number of PC visits and a higher rate of hospital admissions compared to patients who do not have alcohol dependence [16,17]. Furthermore, patients with hazardous alcohol consumption have a greater use of emergency departments and hospital services compared to abstainers [19]. However, there are national and international publications indicating that increased alcohol consumption in the general population is associated with a decreased use of health services [20,21,22]. A national study carried out in a Spanish population aged 16 years and older revealed a negative dose–response relationship between alcohol consumption and the use of hospital and outpatient services [20]. Furthermore, a US study piloted in the adult population indicated that individuals who drink alcohol used emergency services and PC less compared to abstainers [21].

From the public health perspective, the increased use of health services among university students with hazardous alcohol consumption or binge drinking is a relevant issue that requires major attention due to the practical implications that it causes [15,16,17,18]. A higher utilization of emergency services or primary care caused by alcohol consumption implies a greater demand for health care professionals and an increase in costs in the health system that needs to be addressed by policymakers. Since clinical settings are important sites for the identification of unhealthy alcohol use and for the initiation of brief interventions, the evaluation of the pattern of alcohol consumption and the use of health services could provide relevant information to address this public health issue.

Taking the health consequences created by alcohol consumption in young adults into consideration and the existing mixed evidence on the use of healthcare resources in university students, the current study presents the hypothesis that hazardous alcohol consumption and binge drinking patterns in university students are associated with greater use of emergency services and PC.

The aim of the study was to examine the association between alcohol consumption patterns (hazardous alcohol consumption and binge drinking) and the use of emergency services and PC consultations in first-year university students.

## 2. Materials and Methods

### 2.1. Study Design and Sample

This study is an observational, cross-sectional analysis of a dynamic cohort of first-year university students belonging to one of the following Spanish universities: León, Cantabria, Jaén, Vigo, Granada, Huelva, Salamanca, Valladolid, Alicante, and Valencia, all of which are part of the uniHcos project (University, Lifestyles, Follow-up Cohort). The uniHcos project is a multicenter study designed to examine the habits and lifestyles of Spanish university students [23]. The study received approval from the Ethics Committee of the University of León.

Selection criteria: (1) To be a first-year university student enrolled in a Spanish university included in the uniHcos project; (2) to complete the self-administered form and grant informed consent for participation in the study.

Since the uniHcos project is a dynamic cohort, we did not establish a minimum sample size for this study.

### 2.2. Data Collection

Participants were recruited through their university account email. The email included information on the objectives of the uniHcos project and a link to the mandatory informed consent form that had to be completed prior to answering the study questionnaire. The students who agreed to participate completed the self-reported online questionnaire between October 2011 and March 2018 using the SphinxOnline^®^ platform (Le Shphinx Developpement, Chavanod, France). The questionnaire included questions on alcohol consumption from the National Health Survey (ENS) [24] and the Survey on Alcohol and Drugs in Spain (EDADES) [4].

Two patterns of alcohol consumption were studied: hazardous alcohol consumption and binge drinking. Both patterns were calculated using the Alcohol-Dependent Disorders Identification Test (AUDIT) questionnaire [25], which was validated in this population by Kokotailo et al. [26] and Verhoog et al. [27]. A positive result or hazardous alcohol use is considered to be an AUDIT score ≥ 8. This score determines the risk of developing alcohol consumption problems. Binge drinking (BD) or heavy episodic consumption was defined as ingesting six or more alcoholic beverages in a single session, both for men and women [25].

The use of emergency services was assessed by the question: “Have you used an emergency service because of a problem or illness in the last 12 months?” and use of a PC service was assessed by the question: “Have you consulted with a family doctor in the last four weeks for any problem, discomfort, or illness?” The answer to both questions could be “yes” or “no”. Additionally, the number of visits to the emergency department was quantified in three categories: none, one visit, two or more visits.

### 2.3. Data Analysis

A descriptive analysis was performed where central tendency measurements (mean and median) and dispersion (standard deviation and range) of the quantitative variables were calculated, as well as the prevalence of the qualitative variables.

To evaluate the relationship between the use of health services and dependent variables (hazardous alcohol consumption and binge drinking), we used the chi-squared test and Student’s *t*-test, as well as unconditional logistic regression analysis to calculate the odds ratio (OR) and 95% confidence interval (CI). All models were stratified by sex and adjusted for age, occupation, university degree, and type of residence. All statistical analyses were performed using IBM-SPSS statistical package, version 20.0 (IBM Corp. Released 2011. IBM SPSS Statistics for Windows, Version 20.0. Armonk, NY, USA) with a significance level of 95% (*p* = 0.05).

## 3. Results

There were 10,167 participants who completed the questionnaire, 72.2% (95% CI: 70.90–77.2) of whom were women. The participants’ age ranged between 17 and 63 years (mean: 22.11 years; SD: 4.51; 95% CI: 20.02–20.20). Full-time students represented 66.2% (95% CI: 65.3–67.2) of the respondents, 10.7% (95% CI: 10.1–11.3) combined studying and work, and 23.1% (95% CI: 22.3–24.0) were students and looking for work; 39.9% (95% CI: 39.0–40.9) of the respondents were enrolled in a degree in Social and Legal Sciences, 22.3% (95% CI: 21.5–23.1)—in Health Sciences, 15.3%—in Sciences, 12%—in Arts and Humanities, and 10% (95% CI: 11.4–12.7)—in Engineering and Architecture (Table 1).

The prevalence of hazardous alcohol use in the surveyed population was 16.9% (95% CI: 16.2–17.6), while the prevalence of binge drinking was 48.8% (95% CI: 47.9–49.8).

Table 2 shows the participants’ hazardous alcohol use and binge drinking pattern depending on sociodemographic and academic variables. In terms of hazardous alcohol use, significant differences were observed related to age (*p* < 0.001; higher percentage of hazardous alcohol use in the students aged 17–20 years), sex (*p* < 0.001; higher hazardous alcohol consumption in women), place of residence (*p* < 0.001; higher consumption in those surveyed who rented), and university degree (*p* < 0.001; higher consumption in students enrolled in the field of Social and Legal Sciences).

The prevalence of those surveyed with hazardous alcohol use who had two or more visits to emergency services and PC was 21.7% (95% CI: 19.9–23.5) and 15.9% (95% CI: 13.4–18.4), respectively, while the prevalence of individuals with a binge drinking pattern who consulted emergency services and PC was 55.3% (95% CI: 54.5–59.0) and 48.5% (95% CI: 45.0–51.9), respectively. Table 3 shows the prevalence of emergency services and PC use in terms of alcohol consumption patterns. There were significant differences in emergency department visits between the participants surveyed who had a hazardous drinking pattern (*p* < 0.001) and those with a binge drinking pattern (*p* < 0.001).

Table 4 shows the prevalence of emergency services and PC use in terms of alcohol consumption patterns stratified by sex. For both sexes, there were significant differences in emergency department visits between the participants surveyed who had a hazardous drinking pattern (*p* < 0.001) and those with a binge drinking pattern (*p* < 0.001).

Table 5 shows the results of the logistic regression model for visits to the emergency department and PC, with statistically significant values in the hazardous alcohol consumption and binge drinking patterns and visits to the emergency department (*p* < 0.001 in both instances).

Regarding the factors associated with the use of health services according to the pattern of alcohol consumption, we found that sex was associated with visits to the emergency department and primary care consultations in both patterns of alcohol use (fewer visits to both health services in men who had hazardous alcohol use or binge drinking) (Table 6). Furthermore, there was a significant association between the type of residence and the use of health services (more visits to the emergency department and primary care among those who had binge drinking and lived in a rented apartment).

## 4. Discussion

The existing literature on the university students’ lifestyle indicates alcohol as one of the main public health problems in this population group [28,29]. Although a slight decrease in alcohol consumption has been observed among young adults since 2015, hazardous alcohol consumption persists among university students, and binge drinking has increased in the last decade, as well as the use of health services [4]. This study aimed to analyze the association between alcohol consumption patterns and the use of emergency services and PC by university students. The results reveal that the group of students enrolled in the university with hazardous alcohol use or binge drinking has a greater use of emergency services. However, no significant association was observed between these consumption patterns and PC visits.

The end of adolescence and the beginning of adulthood are marked by various changes of a social, educational, occupational, and economic nature [30]. The change of address, the beginning of university studies, first employment contracts, or new social circles characterize this vital period, which can be associated with a higher level of stress and frustration that can lead to unhealthy behaviors, such as alcohol consumption [6]. Similarly, in public health, behavioral aspects related to social structures typical of these ages are related. Young university students consider it more important to belong to a higher social group than the meaning of a more static risk, where social interaction networks play an essential role in addictions [31].

Current evidence indicates that hazardous alcohol consumption and binge drinking are higher in adolescence and early adulthood [4]. Despite being the age group with significant alcohol consumption, there is considerable variation in the prevalence of hazardous and binge drinking recorded in international texts. The prevalence of binge drinking varies in the literature from 24% to 64% in men and from 14% to 63% in women [32,33,34]. The heavy episodic consumption registered in this study is in line with international studies that have reflected a high percentage of individuals with binge drinking. The study by Beenstock et al. [35], carried out in 2008, identified the pattern of hazardous alcohol consumption in university students studying Humanities and Social Sciences, Agriculture, and Engineering from the United Kingdom in 82% of those surveyed. Another study carried out between 2008–2009 in Science, Sport Science, and Art students in the United Kingdom identified hazardous consumption of alcohol at 46% [36]. Conversely, an investigation carried out in New Zealand by Kypri et al., in which the prevalence of hazardous alcohol consumption was compared among university students and non-university young people aged 18 to 23, revealed that the prevalence of risky alcohol consumption in university students was almost double that of non-university students, while harmful alcohol consumption (AUDIT ≥ 15) was triple in university students compared to non-university students [37]. The previous findings show that the variations in prevalence of alcohol consumption may reflect differences in occupation (being a full-time student, employed, or both) of the different samples, as well as cultural differences in alcohol consumption worldwide.

A considerable amount of research in health services is focused on patients with alcohol intoxication, a consequence of ingesting large amounts of alcohol, and their use of emergency services [38,39,40]. Data from the 2005 and 2010 U.S. National Alcohol Surveys conducted in the general population do not show significant differences between patients with hazardous alcohol consumption and patients without hazardous consumption in terms of use of emergency services, PC, and hospitalizations in the previous year [41]. However, patients with alcohol use disorder (defined using the Diagnostic and Statistical Manual, 4th revision (DSM-IV), criteria) were significantly more likely to visit the emergency room in the previous year (18.2% vs. 11.6%; *p* = 0.003), report more PC visits (*p* = 0.05), and some hospitalizations (11.2% vs. 6.7%; *p* = 0.019) compared to individuals who did not present this disorder [40]. Similarly, our findings are in line with the results provided by Miquel et al. from a sample of more than 606,948 patients in Catalonia, where a greater use of emergency services was detected in individuals with high-risk alcohol use, but no greater use of PC services [42].

Screening for alcohol use in college students and using health services can help identify problematic drinking behaviors [43] and recognize groups of students at higher risk who may later be the target of intervention techniques [44], as well as develop personalized strategies to reduce the risk of alcohol consumption [45,46] as per the recommendations from the European Observatory on Drugs [47] and the United Nations Office in Drugs and Crime (UNOCD) and the World Health Organization [48]. In a study carried out in Australia, a web page was developed that had a series of online tools, one of which was the AUDIT questionnaire, that individually recorded hazardous alcohol consumption in university students. Guidelines were offered to reduce alcohol drinking, and information on the emotional and behavioral consequences of hazardous alcohol consumption and its relation to traffic accidents, estimates of the students’ annual alcohol expenditure, and links to useful information to stop drinking and smoking were also included. After these interventions, the study group exhibited a reduction of 17% in alcohol consumption after one month, compared to the control group, and 11% after 6 months. These differences were mainly due to fewer episodes of alcohol consumption and, to a lesser extent, less alcohol consumed in each episode [49].

Among the limitations of the current study it is necessary to highlight that the questionnaire used was not validated, though it does include validated questions from national questionnaires, like the EDADES Survey [4] and AUDIT [25]. Another limitation of the study lies in declaring alcohol consumption since consumption was registered through self-reporting by the respondents. Knowing they were being surveyed could lead to underestimation of the prevalence of participants’ alcohol consumption, but there are studies that consider it valid and reliable [50]. It is also worthy of mentioning that the frequency of use of emergency services or PC is correlational and not causative in the dataset. Additionally, it is necessary to highlight that “age” was not included as an inclusion–exclusion criterion. Since university is not compulsory, first-year university students were not necessarily young adults, and this aspect should be included as a limitation of the study.

The results of this study can serve as a basis in developing alcohol consumption prevention programs in university students focused on brief intervention and referral to treatment, or recovery. Various investigations have raised the need to carry out public health actions to reduce alcohol consumption in university students. These recommendations are based on reducing the number of alcohol outlets around college campuses, raising prices through taxes, enforcing laws related to alcohol consumption, restrictions on alcohol advertising, increasing the minimum age of purchase to age 20 or 21, stricter control of alcohol sales at university events, and screening and interventions in health services attended by university students [37,47,48,51]. These constraints are contemplated within the framework of actions to promote health and the Spanish Network of Healthy Universities (REUS) [52] and are part of international strategies like the Global Strategy to reduce the harmful use of alcohol (2010) [53] and the World Health Organization’s SAFER initiative (2018) [54], the strategy to help European Union countries reduce alcohol-related harm [55] and goal 3.5 of the 2030 United Nations Agenda [56], as well as the National Strategy on Addictions 2017–2024 [57] and the Strategy for Health Promotion and Prevention in the National Health System [58].

As such, multicenter longitudinal studies are needed to evaluate the association of hazardous alcohol use and binge drinking and the use of health services in the university setting [59], as well as longitudinal analyses that examine other risk factors in this population group, like tobacco or other drugs, and their possible impact on health services.

## 5. Conclusions

The current study shows that an association exists between university students with a pattern of hazardous alcohol consumption or binge drinking and greater use of emergency services. However, no significant association is observed between the said consumption patterns and attendance at a PC consultation. As such, it is necessary to implement alcohol consumption prevention programs in universities to reduce binge drinking and hazardous alcohol consumption. These strategies can be implemented from the academic field, for example, at PC consultations.

## Figures and Tables

**Table 1 ijerph-19-06158-t001:** Main characteristics of the study sample (*n* = 10,167).

Main Characteristics of the Study Sample		*n* (%)	95% CI
**Sex**	Men	2823 (27.8)	25.6–27.3
	Women	7344 (72.2)	67.9–69.7
**Age**	17–20	7810 (76.8)	76.0–77.6
	21–24	1496 (14.7)	14.0–15.4
	≥25	861 (8.5)	7.9–9.0
**Occupation**	Student	6730 (66.2)	65.3–67.1
	Student and employee	1085 (10.7)	10.1–11.3
	Student and looking for work	2352 (23.1)	22.3–24.0
**Residence**	University residence	1297 (12.8)	12.1–13.4
	Family/own home	4877 (48.0)	47.0–48.9
	Rental apartment	3993 (39.3)	38.3–40.2
**Degree**	Arts and Humanities	1225 (12.0)	11.4–12.7
	Sciences	1556 (15.3)	14.6–16.0
	Health Sciences	2268 (22.3)	21.5–23.1
	Social and Legal Sciences	4061 (39.9)	39.0–40.9
	Engineering and architecture	1057 (10.4)	9.8–11.0
**University**	Alicante	854 (8.4)	7.9–8.9
	Cantabria	88 (0.9)	0.7–1.0
	Castilla La Mancha	192 (1.9)	1.6–2.2
	Granada	2936 (28.9)	28.0–29.8
	Huelva	428 (4.2)	3.8–4.6
	Jaén	290 (2.9)	2.5–3.2
	León	900 (8.9)	8.3–9.4
	Salamanca	1211 (11.9)	11.3–12.5
	Valencia	1452 (14.3)	13.6–15.0
	Valladolid	616 (6.1)	5.6–6.5
	Vigo	1200 (11.8)	11.2–12.4

95% CI = 95% confidence interval.

**Table 2 ijerph-19-06158-t002:** Patterns of alcohol use in university students according to sociodemographic and occupational variables.

Variables		Hazardous Alcohol Use			Binge Drinking		
		No (*n* = 8450) *n* (%)	Yes (*n* = 1717) *n* (%)	*p*	No (*n* = 5203) *n* (%)	Yes (*n* = 4964) *n* (%)	*p*
**Sex**	Men	2188 (25.9)	635 (37.0)	<0.001	1484 (28.5)	1339 (27.0)	0.082
	Women	6262 (74.1)	1082 (63.0)		3719 (71.5)	3625 (73.0)	
**Age**	17–20	6459 (76.4)	1351 (78.7)	<0.001	3969 (76.3)	3841 (77.4)	<0.001
	21–24	1237 (14.6)	259 (15.1)		731 (14.0)	765 (15.4)	
	≥25	754 (8.9)	107 (6.2)		503 (9.7)	358 (7.2)	
**Occupation**	Student	5621 (66.5)	1109 (64.6)	0.077	3511 (67.5)	3219 (64.8)	0.012
	Student and employee	910 (10.8)	175 (10.2)		548 (10.5)	537 (10.8)	
	Student and looking for work	1919 (22.7)	433 (25.2)		1144 (22.0)	1208 (24.3)	
**Residence**	University residence	1034 (12.2)	263 (15.3)	<0.001	613 (11.8)	684 (13.8)	<0.001
	Family/own home	4240 (50.2)	637 (37.1)		2833 (54.4)	2044 (41.2)	
	Rental apartment	3176 (37.6)	817 (47.6)		1757 (33.8)	2236 (45.0)	
**Degree**	Arts and Humanities	1019 (12.1)	206 (12.0)	<0.001	650 (12.5)	575 (11.6)	<0.001
	Sciences	1298 (15.4)	258 (15.0)		841 (16.2)	715 (14.4)	
	Health Sciences	1956 (23.1)	312 (18.2)		1203 (23.1)	1065 (21.5)	
	Social and Legal Sciences	3304 (39.1)	757 (44.1)		1936 (37.2)	2125 (42.8)	
	Engineering and architecture	873 (10.3)	184 (10.7)		573 (11.0)	484 (9.8)	

**Table 3 ijerph-19-06158-t003:** Health services use among university students according to the pattern of alcohol use.

Pattern of Alcohol Use		Health Services Use							
		Emergency Department				Primary Care			
		No (*n* = 6380) *n* (%)	Yes (*n* = 3787) *n* (%)		*p*	No (*n* = 7592) *n* (%)	Yes (*n* = 2575) *n* (%)		*p*
**Hazardous alcohol use**	No	5413 (84.8)	967 (80.2)		<0.001	6315 (83.2)	1277 (82.9)		0.367
	Yes	3037 (15.2)	2030 (19.8)			2135 (16.8)	3625 (17.1)		
**Binge drinking**	No	3446 (54.0)	2934 (46.4)		<0.001	3905 (51.4)	3687 (50.4)		0.755
	Yes	1757 (46.0)	2030 (53.6)			1298 (48.6)	1277 (49.6)		
		**Number of visits to the emergency department**				**Number of visits to primary care**			
		**0** ***n* (%)**	**1** ***n* (%)**	**≥2** ***n* (%)**	** *p* **	**0** ***n* (%)**	**1** ***n* (%)**	**≥2** ***n* (%)**	** *p* **
**Hazardous alcohol use**	No	5417 (84.8)	1486 (82.3)	1547 (78.3)	<0.001	6316 (83.2)	1447 (82.4)	687 (84.1)	0.523
	Yes	969 (15.2)	319 (17.7)	429 (21.7)		1277 (16.8)	310 (17.6)	130 (15.9)	
**Binge drinking**	No	3448 (54.0)	872 (48.3)	833 (44.7)	<0.001	3905 (51.4)	877 (16.9)	421 (8.1)	0.508
	Yes	2938 (46.0)	933 (51.7)	1093 (55.3)		3688 (48.6)	880 (50.1)	396 (48.5)	

**Table 4 ijerph-19-06158-t004:** Health services use among university students according to the pattern of alcohol use, stratified by sex.

Patterns of Alcohol Use	Visits to Primary Care						
	Women				Men		
		No (*n* = 5315) *n* (%)	Yes (*n* = 2029) *n* (%)	*p*	No (*n* = 2277) *n* (%)	Yes (*n* = 546) *n* (%)	*p*
Hazardous alcohol use	No	4539 (85.4)	1723 (84.9)	0.606	1776 (78.0)	412 (75.5)	0.704
	Yes	776 (14.6)	306 (15.1)		501 (22.0)	134 (24.5)	
Binge drinking	No	2712 (51.0)	1007 (49.6)	0.285	1193 (52.4)	291(53.2)	0.202
	Yes	2603 (49.0)	1022 (50.4)		1084 (47.6)	255 (46.8)	
**Patterns of Alcohol Use**	**Visits to the emergency department**						
	**Women**				**Men**		
		**No (*n* = 4469)** ***n* (%)**	**Yes (*n* = 2875)** ***n* (%)**	** *p* **	**No (*n* = 1911)** ***n* (%)**	**Yes (*n* = 912)** ***n* (%)**	** *p* **
Hazardous alcohol use	No	3880 (86.8)	2382 (82.8)	<0.001	1533 (80.2)	655 (71.8)	<0.001
	Yes	589 (13.2)	493 (17.2)		378 (19.8)	257 (28.2)	
Binge drinking	No	2383 (53.3)	1336 (46.5)	<0.001	1063 (55.6)	421 (46.2)	<0.001
	Yes	2086 (46.7)	1539 (53.5)		848 (44.4)	491(53.8)	

**Table 5 ijerph-19-06158-t005:** Association between the patterns of alcohol consumption and use of health services: logistic regression model.

Visits to Primary Care			
Patterns of Alcohol Use	aOR *	95% CI	*p*
Hazardous alcohol use	1.07	0.95–1.21	0.246
Binge drinking	1.03	0.94–1.13	0.521
**Visits to the emergency department**			
**Patterns of alcohol use**	**aOR ***	**95% CI**	** *p* **
Hazardous alcohol use	1.38	1.24–1.53	<0.001
Binge drinking	1.30	1.19–1.40	<0.001

aOR = adjusted odds ratio; 95% CI = 95% confidence interval. * Adjusted by age, sex, residence, occupation, degree, and pattern of alcohol consumption.

**Table 6 ijerph-19-06158-t006:** Factors associated with the use of health services according to the pattern of alcohol consumption: logistic regression model.

Visits to the Emergency Department								Visits to Primary Care					
Variables		Hazardous Drinking			Binge Drinking			Hazardous Drinking			Binge Drinking		
		OR *	95% CI	*p*	OR *	95% CI	*p*	OR *	95% CI	*p*	OR *	95% CI	*p*
Sex	Men	0.74	0.68–0.82	<0.001	0.77	0.70–0.85	<0.001	0.64	0.57–0.72	<0.001	0.65	0.58–0.72	<0.001
	Women	1			1			1			1		
Age	17–20	1			1			1			1		
	21–24	0.98	0.83–1.15	0.806	0.98	0.83–1.14	0.762	1.08	0.91–1.29	0.366	1.09	0.91–1.29	0.357
	≥25	1.10	0.84–1.20	0.948	0.99	0.83–1.19	0.986	1.17	0.96–1.42	0.120	1.17	0.96–1.42	0.120
Occupation	Student	1.10	0.88–1.20	0.680	1.025	0.88–1.19	0.881	1.19	1.02–1.41	0.044	1.20	1.02–1.40	0.030
	Student and employee	0.81	0.73–0.89	<0.001	0.81	0.73–0.89	0.730	0.74	0.66–0.83	0.590	0.74	0.67–0.83	<0.001
	Student and looking for work	1			1			1			1		
Residence	University residence	0.77	0.71–0.84	0.706	0.78	0.71–0.85		1.04	0.89–1.22	0.590	1.04	0.89–1.22	0.578
	Family/own home	1			1		<0.001	1			1		
	Rental apartment	1.08	0.94–1.23	0.942	1.09	0.95–1.25	<0.001	1.11	1.01–1.22	0.044	1.10	1.01–1.22	0.048
Degree	Arts and Humanities	1.11	0.97–1.26	0.123	0.88	0.75–1.03	0.115	0.91	0.76–1.09	0.312	0.91	0.76–1.09	0.311
	Sciences	1			1			1			1		
	Health Sciences	1.15	0.99–1.33	0.056	1.14	0.98–1.31	0.080	1.07	0.91–1.26	0.394	1.07	0.91–1.26	0.408
	Social and Legal Sciences	1.12	0.98–1.28	0.097	1.11	0.97–1.27	0.127	1.09	0.94–1.27	0.225	1.10	0.94–1.27	0.226
	Engineering and architecture	1.03	0.86–1.23	0.718	1.02	0.57–1.22	0.793	0.89	0.73–1.10	0.284	0.89	0.73–1.09	0.277

aOR = adjusted odds ratio; 95% CI = 95% confidence interval. * Adjusted by age, sex, residence, occupation, degree, and pattern of alcohol consumption.

## Data Availability

The datasets analyzed in the current study are available from the corresponding author on reasonable request.

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
