# Peer review of "Patterns of Alcohol Consumption and Use of Health Services in Spanish University Students: UniHcos Project"

_ijerph, 2022, doi:10.3390/ijerph19106158_

Round 1
Reviewer 1 Report
This study aimed to analyze the association between alcohol consumption patterns and the use of emergency services and PC by university students. The findings may help develop prevention programs for university students. There are several comments I raise as follows:
- Only first-year university students were enrolled in the study, a selection bias may limit the findings.
- Women are the majority in the study, a separate analysis of gender is suggested to find out the different patterns between the two groups.
- Using a logistic regression model, a full model result that includes each adjusted OR of variables such as age, sex, residence, occupation, and degree will be able to extend the findings more than just presenting them with the patterns of alcohol consumption.
Reviewer 2 Report
Thank you for the opportunity to review this article.
The authors investigate the association between patterns of alcohol consumption and use of health services among University students.
I think the study is very simple, the authors provide little investigation about possible mediating factors, therefore the study results are low informative and scientific soundness of the paper, as presented, is limited.
I report below some more considerations.
Line 64: “at the national level”. Which nation? For clarity, the authors should specify the country they refer to.
From the introduction section, it is not clear what is the relevance of the paper’s topic. That is, why would it be a relevant issue investigating the association between patterns of alcohol consumption and use of emergency care and PC? What are the practical and beneficial implications? The authors should provide this information in the Introduction section.
Line 104-106: I suggest to shift the aim of the study in the Introduction section, before the research Hypothesis.
Line 115-117: Weren’t there “age” inclusion-exclusion criteria? As University is not compulsory school, first-year University student is not necessarily young-adult. The authors should indicate this aspect in the Limitation section.
Line 136-141: On which basis the authors chose 12 month and 4 weeks as time indicators?
Line 269: “The results of this study can serve as a basis in developing alcohol consumption prevention programs in university students.” Can the authors provide some examples of how the manuscript’s results can contribute? What about practical implications?
The study provides limited discussion about the results of the present research. The authors provide a contextual framework of the literature on this topic, however there is not a specific discussion on the specific results of the study.
Author Response
Reviewer 2: The authors investigate the association between patterns of alcohol consumption and use of health services among University students.
I think the study is very simple, the authors provide little investigation about possible mediating factors, therefore the study results are low informative and scientific soundness of the paper, as presented, is limited.
I report below some more considerations.
Line 64: “at the national level”. Which nation? For clarity, the authors should specify the country they refer to.
This assumption has been clarified in the Introduction section.
From the introduction section, it is not clear what is the relevance of the paper’s topic. That is, why would it be a relevant issue investigating the association between patterns of alcohol consumption and use of emergency care and PC? What are the practical and beneficial implications? The authors should provide this information in the Introduction section.
Thank you for the thoughtful comment. Following the reviewer´s recommendation, this point has been clarified in the Introduction.
Attached is the paragraph included in the Introduction section:
¨From a public health perspective, the increased use of health services among university students with hazardous alcohol consumption or binge drinking is a relevant issue that requires a major attention due to the practical implications that it causes [15-18]. A higher utilization of emergency services or primary care caused by alcohol consumption implies a greater demand for health care professionals and an increase of costs in the health system that needs to be addressed by policymakers. Since the clinical settings are important sites for the identification of unhealthy alcohol use and for the initiation of brief intervention, the evaluation of the pattern of alcohol consumption and the use of health services could provide relevant information to address this public health issue.¨
Line 104-106: I suggest to shift the aim of the study in the Introduction section, before the research Hypothesis.
We appreciate the reviewer´s suggestion. We have shifted the aim of the study in the Introduction section.
Line 115-117: Weren’t there “age” inclusion-exclusion criteria? As University is not compulsory school, first-year University student is not necessarily young-adult. The authors should indicate this aspect in the Limitation section.
Thank you for the thoughtful comment. We added the reviewer´s suggestion to the limitation section:
¨Additionally, it is necessary to highlight that ¨age¨ was not included as an inclusion-exclusion criteria. Since University is not compulsory, first-year University students were not necessarily young adult and this aspect should be included as a limitation of the study.¨
Line 136-141: On which basis the authors chose 12 month and 4 weeks as time indicators?
Since the time indicator used in the AUDIT questionnaire was 12 months, we used the same time indicator to evaluate the visits to the emergency services.
Regarding the primary care consultations, the time indicator was 4 weeks as the frequency of visits to primary healthcare are more prevalent (see Table 2 of the following article: https://scielo.isciii.es/scielo.php?script=sci_arttext&pid=S1699-695X2019000200050).
Line 269: “The results of this study can serve as a basis in developing alcohol consumption prevention programs in university students.” Can the authors provide some examples of how the manuscript’s results can contribute? What about practical implications?
Prevention programs focused on brief Intervention and referral to treatment, or recovery.
Attached is the information included in the discussion:
¨ The results of this study can serve as a basis in developing alcohol consumption prevention programs in university students focused on brief Intervention and referral to treatment, or recovery.¨

Reviewer 3 Report
Thank you for the opportunity to review this very useful and well designed study. I have really only a few observations that are quite minor
In line 157-160 - is this demographic spread by program typical of the university population throughout these Spanish institutions?
In Table 3 -Patterns of alcohol use - n% number seems to be missing
In terms of limitations, it is valid to recognize that frequency of use of ER or PC is correlational and not causative in the data set
Author Response
Reviewer 3: The study provides limited discussion about the results of the present research. The authors provide a contextual framework of the literature on this topic, however there is not a specific discussion on the specific results of the study.
Thank you for the opportunity to review this very useful and well-designed study. I have really only a few observations that are quite minor
In line 157-160 - is this demographic spread by program typical of the university population throughout these Spanish institutions?
Thank you for the thoughtful comment. The demographic results obtained in our study are similar to the results obtained in the Spanish university population. Attached are the main findings from the last National Survey of the Spanish university population: chrome-extension://efaidnbmnnnibpcajpcglclefindmkaj/viewer.html?pdfurl=https%3A%2F%2Fwww.universidades.gob.es%2Fstfls%2Funiversidades%2FEstadisticas%2Fficheros%2FPpalesResulEEU.pdf&clen=486636&chunk=true
In Table 3 -Patterns of alcohol use - n% number seems to be missing
This information has been clarified in the table.
In terms of limitations, it is valid to recognize that frequency of use of ER or PC is correlational and not causative in the data set
This assumption has been added to the discussion.

Round 2
Reviewer 2 Report
The revised version in now suitable for publication